# Comparison on Major Gene Mutations Related to Rifampicin and Isoniazid Resistance between Beijing and Non-Beijing Strains of *Mycobacterium tuberculosis*: A Systematic Review and Bayesian Meta-Analysis

**DOI:** 10.3390/genes13101849

**Published:** 2022-10-13

**Authors:** Shengqiong Guo, Virasakdi Chongsuvivatwong, Shiguang Lei

**Affiliations:** 1Guizhou Provincial Center for Disease Prevention and Control, Guiyang 550004, China; 2Department of Epidemiology, Faculty of Medicine, Prince of Songkla University, Hat Yai 90110, Thailand

**Keywords:** *Mycobacterium tuberculosis*, Beijing and non-Beijing strain, mutation of gene, MDR, rifampicin and isoniazid

## Abstract

**Objective:** The Beijing strain of *Mycobacterium tuberculosis* (MTB) is controversially presented as the predominant genotype and is more drug resistant to rifampicin and isoniazid compared to the non-Beijing strain. We aimed to compare the major gene mutations related to rifampicin and isoniazid drug resistance between Beijing and non-Beijing genotypes, and to extract the best evidence using the evidence-based methods for improving the service of TB control programs based on genetics of MTB. **Method:** Literature was searched in Google Scholar, PubMed and CNKI Database. Data analysis was conducted in R software. The conventional and Bayesian random-effects models were employed for meta-analysis, combining the examinations of publication bias and sensitivity. **Results:** Of the 8785 strains in the pooled studies, 5225 were identified as Beijing strains and 3560 as non-Beijing strains. The maximum and minimum strain sizes were 876 and 55, respectively. The mutations prevalence of *rpoB, katG, inhA* and *oxyR-ahpC* in Beijing strains was 52.40% (2738/5225), 57.88% (2781/4805), 12.75% (454/3562) and 6.26% (108/1724), respectively, and that in non-Beijing strains was 26.12% (930/3560), 28.65% (834/2911), 10.67% (157/1472) and 7.21% (33/458), separately. The pooled posterior value of OR for the mutations of *rpoB* was 2.72 ((95% confidence interval (CI): 1.90, 3.94) times higher in Beijing than in non-Beijing strains. That value for *katG* was 3.22 (95% CI: 2.12, 4.90) times. The estimate for *inhA* was 1.41 (95% CI: 0.97, 2.08) times higher in the non-Beijing than in Beijing strains. That for *oxyR-ahpC* was 1.46 (95% CI: 0.87, 2.48) times. The principal patterns of the variants for the mutations of the four genes were *rpoB* S531L, *katG* S315T, *inhA*-15C > T and *oxyR-ahpC* intergenic region. **Conclusion:** The mutations in *rpoB* and *katG* genes in Beijing are significantly more common than that in non-Beijing strains of MTB. We do not have sufficient evidence to support that the prevalence of mutations of *inhA* and *oxyR-ahpC* is higher in non-Beijing than in Beijing strains, which provides a reference basis for clinical medication selection.

## 1. Introduction

Tuberculosis (TB) is one of the deadliest transmissible diseases that cause death worldwide. However, only 10% of people infected with *Mycobacterium tuberculosis* (MTB) develop TB disease [1], indicating that either the host or the pathogen’s genetic factors may play a critical role in determining the occurrence of TB disease. The Beijing strain of MTB is presented as the predominant strain. It plays a vital role in many countries, such as Bangladesh [2], Upper Myanmar [3] and China [4,5], with the Beijing strain accounting for 26.8%, 71.4% and 81.7%, respectively. The latter country, China, holds the second highest tuberculosis (TB) burden, presenting 8.5% of case notifications worldwide [6].

The Beijing strain of MTB is reported to be more virulent, more pathogenic, and faster-growing, with more histopathological changes and drug resistance, especially multidrug-resistance TB (MDR-TB) tendencies, than other strains, leading to a higher mortality rate [7]. The rate of treatment success in MDR-TB remains low, reaching only 47–62.7% [2,8]. MDR TB is not only a severe clinical and epidemiological problem but also entails substantial economic costs of management.

Thus, treating patients with resistance to the main anti-TB agents, such as rifampicin (RIF) and isoniazid (INH), may be many times more expensive compared to treatment costs incurred by the management of TB susceptible to the main medication panel [9]. MDR-TB poses a significant threat not only to the individual faced with diminished chances of cure compared with non-MDR-TB, but also to the community, as outbreaks of MDR-TB have been shown to have devastating consequences [10].

Furthermore, some studies have suggested an association between drug resistance and some MTB genotypes [11,12,13]. Resistance to anti-TB drugs in MTB mainly arises from genomic mutations in genes encoding either the drug target or enzymes involved in drug activation [14,15]. Even some efflux pump genes, such as *drrA, drrB, efpA, Rv2459, Rv1634, and Rv1250* [16], were also reported to be related to the resistance of MDR; however, some previous studies suggested the more common candidate genes’ mutations to be related to MDR [10,17,18,19], such as the *rpoB* gene is associated with rifampicin, and *katG*, *inhA*, and *ahpC* genes are related to isoniazid resistance [10]. Other genes mutations related to drug resistance are also reported, such as *rpsL* K43R to streptomycin, *embB* M306V to Ethambutol, *pncA* promoter T (-11) C to pyrazinamide, *gyrA* A90V to fluoroquinolones, *RRS* A1401G to second-line injection drug, and *fabGl*_promoter C(-15) T to Ethionamide) [20].

It is addressed that 95% of rifampicin resistance (RR) is associated with the mutation in the 81 bp rifampicin resistance determining region (RRDR) [21]. Resistance mutations in RRDR of the *rpoB* gene were found to be associated with phenotypic RIF resistance. [22]. The *rpoB* gene codes the β-subunit of DNA-dependent RNA-polymerase, which acts as a major target for RIF, and up to 95–98% of RIF-resistance strains exhibit mutations in the *rpoB* gene, whereas 90–95% of these mutations are located in RRDR [8,23].

INH resistance is associated with mutations in multiple loci, such as the catalase-peroxidase gene (*katG*), the enoyl-ACP reductase gene (*inhA*) and its promoter, the alkyl hydroperoxide reductase gene (*ahpC*), and the intergenic region between the *oxyR* and *ahpC* (*oxyR-ahpC*) genes, which is distinguished from that of RIF [24,25,26]. One specific *KatG* variant, S315T, is found in 94% of INH-resistance clinical isolates. Around 15 mutations in *inhA* have been identified in INH-resistance clinical isolates, although two of them were also found in INH-sensitive strains. In this regard, the analysis of gene expression profiling of the Beijing strain of MTB can give us a snapshot of actively expressed genes under various conditions, even though some other researchers hold the opposite issue [27].

Due to the discrepancies between studies possibly resulting from the small sample sizes and variant detection methods of genes in different areas, pooled evidence is needed to provide better evidence that inform policymakers’ decisions for controlling TB. Bayesian meta-analysis (BMA) is reported that it harbors more robustness [28] than the conventional meta-analysis (MA) and is not limited to the premise of classical statistical methods, which can be combined with a priori information, sample information and general information, can obtain the posterior distribution easily and is based on its effect quantity variance between the mergers of the values, research, other parameters, and 95% CI, e.g., the shrinkage estimation values with the consideration of the potential publication biases. It is believed that Bayesian statistical methods will be more widely used in evidence-based medicine/meta-analyses [28].

This systematic review focused on combining the results about genes relevant to MDR with the concepts of the classifications of Beijing and non-Beijing using conventional meta-analysis MA and BMA. We aimed to compare the major gene mutations related to RIF and INH resistance between Beijing and non-Beijing genotypes and extract the best evidence using evidence-based methods for improving the TB control program’s service based on the genetics of MTB.

## 2. Methods

### 2.1. Study Design

This systematic review and Bayesian meta-analysis were conducted according to the Preferred Reporting Items for Systematic Reviews and Meta-Analyses Statement (PRISMA, http://links.lww.com/SLA/C529, accessed on 5 September 2022) (Appendix A) [29] and the meta-analyses of observational studies in epidemiology guidelines [30,31]. Bayesian meta-analysis is performed using Bayesian methods, which provide a profitable opportunity for flexible modeling of inter-study heterogeneity by mildly regularizing priors to obtain a stable estimation, which frequency models prove impossible to calculate [32,33].

### 2.2. Literature Search Strategy

To ensure that a piece of relevant contemporary information was obtained [31], limits were applied to years 1960 onward and MTB genetics or clinic research related to MDR, or RIF/INH drug resistance. Eventually, a retrieval of literature relating the genetics from 1 January 1960 to the present was performed.

Search engines: Google Scholar, PubMed, ResearchGate, ResearchGate, Cochrane Library and Chinese National Knowledge Infrastructure (CNKI) Database.

Search terms: MTB AND Beijing AND non-Beijing AND gene mutation AND MDR, or RIF, or INH drug resistance; *rpoB* mutation AND Beijing AND non-Beijing AND MDR, or RIF; *katG* mutation AND Beijing AND non-Beijing AND MDR, or INH; *inhA* mutation AND Beijing AND non-Beijing AND MDR, or INH; *oxyR-ahpC* mutation AND Beijing AND non-Beijing AND MDR, or INH.

### 2.3. Study Selection Criteria

Inclusion criteria: (1) Full article, abstract, letter presenting the major gene mutations related to MDR of MTB classified as Beijing and non-Beijing strains written in English or Chinese; and (2) Gray literature related to the first point above, which is a kind of information produced outside of traditional publishing and distribution channels, and can include reports, policy literature, working papers, newsletters, government documents, speeches, white papers, urban plans, and so on, written in English or Chinese [34].

Exclusion criteria: (1) Studies only related to the genes of MTB produced by the contacts of the studied subjects or produced by the same subject but obtained through follow-up; (2) studies with drug susceptibility test (DST) involving rifampicin and/or isoniazid only related to children analyzed; and (3) studies conducted in unique sites, such as prisons and asylums.

### 2.4. Data Extraction

Screening of studies and all essential data from the included studies meeting the inclusion criteria were extracted by the investigators (S.G. and V.C.). The principal mutations of the four genes, *rpoB, katG, inhA and oxyR-ahpC*, of MTB related to RIF and INH were input into a predesigned Excel sheet. The results were compared electronically according to the two classification variables, Beijing and not-Beijing strains. The place with a supposed gene absence was labeled with “NA” in the Excel sheet.

The study content recorded the data related to the surname of the first author, country of the subjects, date of publication, study design, sample size, and frequency in the relevant sheet (Appendix A).

Any records with discrepancies were resolved by referring to the source articles. Discrepancies between the two reviewers were resolved by consensus involving all the authors. The R package *metagear* [35,36] was performed for the initial screening articles for the literature review.

### 2.5. Data Synthesis and Statistical Analysis

All analyses were conducted using R software (version 3.6.3) with the following packages, epicalc, medorator and Bayesmeta [28]. The Bayesian random-effects model was used for Bayesian meta-analysis [28,37]. Significant heterogeneity between studies would be considered the presence of heterogeneity when the *p* value is less than 0.05 or I^2^ is greater than 50%.

The leave-one-out [38] and influence sensitivity analyses were also employed by iteratively removing one study at a time while recalculating the odds ratio (OR) to assess the robustness of the pooled values to explore potential sources of inter-study heterogeneity and to further determine the influence of each study, from which the preprint studies had been excluded.

Subgroup analysis was employed for the three groups according to the regions, East Asia, South/Southeast/West/Central Asia and East/North/Central Europe. Potential publication bias was also assessed by the funnel plot, tests of Egger’s liner weighted regression [39] and Begg [40]. Asymmetry of the collected studies’ distribution by visual inspection or *p* value is less than 0.05 was considered as statistically significant [41], indicating the presence of a publication bias evaluated by weight-function. Duval and Tweedie’s trim and fill method’s assumption was considered to reduce the bias in the pooled estimates [42]. To make it more profitable to interpret, logarithms were converted into corresponding constants where appropriate.

## 3. Results

### 3.1. Literature Search Results

In the initial literature search, 1733 relevant articles were identified. After removing 871 duplicates and 573 articles from primary screening, 198 full-text articles were assessed for eligibility in the meta-analysis. Of these, 91 were excluded due to a paucity of sufficient data. Eventually, a total of 134 articles published between 1 January 1960 and 5 March 2022 were included in the quality review part and 31 in the Bayesian meta-analysis part (Figure 1).

### 3.2. Characteristics of Studies Included

As described in Table 1, a total of 31 studies were included in the final Bayesian meta-analysis. The literature has a relatively wide global range covering Asia (China, Korea, Japan, Thailand, Indonesia, Kyrgyzstan, Bangladesh, India, Iran and Turkey) and Europe (Germany, Latvia, Russia, Ukraine and Sweden). Notably, the proportion of studies conducted in China accounted for the majority.

All included studies described critical elements of study design, including study setting, data source, inclusion criteria, participant selection and statistical methods. No studies explained the solution to the missing values, mentioned sample size calculation, or conducted subgroup analysis based on region (Table 1).

### 3.3. Mutations Prevalence for Mutations of Genes

Globally, 8785 pooled MTB isolates were tested to identify MDR-TB, RIF and INH resistance patterns, with 5225 identified as Beijing strains and 3560 as non-Beijing strains. The maximum sample size was 876 strains, and the minimum one was 55 isolates. The prevalence of mutations for *rpoB, katG, inhA* and *oxyR-ahpC* in Beijing strains was 52.40% (2738/5225), 57.88% (2781/4805), 12.75% (454/3562) and 6.26% (108/1724), respectively; and that in non-Beijing strains was 26.12% (930/3560), 28.65% (834/2911), 10.67% (157/1472) and 7.21% (33/458), separately. The principal variants for the four genes were *rpoB* Ser531Leu, *katG* S315T, *inhA*-15C > T and *oxyR-ahpC* intergenic region, respectively (Table 2).

### 3.4. Publication Bias and Sensitivity Analyses

The symmetrical distributions of the funnel plots were detected when the publication biases were evaluated for all the mutations of *rpoB, katG, inhA* and *oxyR-ahpC* among Beijing and non-Beijing strains, paralleled with the *p* > 0.05 of both Egger and Begg tests, indicating the absence of the publication biases. The robustness was detected after sensitivity analysis using leave-one-out and influence tests (Figure 2A–D).

### 3.5. Mutations of Major Genes in Beijing and Non-Beijing Strains

Of the 31 studies, 31 studies were evaluated for the mutations of *rpoB*, 27 studies for the mutations of *katG*, 18 studies for the mutations of *inhA* and 9 studies for that of *oxyR*-*ahpC* [5,10,15,17,20,21,22,24,43,44,45,46,47,48,49,50,51,52,53,54,55,56,57,58,59,60,61,62,63,64,65,66,67,68,69]. The subgroup analysis was conducted for the mutations of *rpoB*, *katG* and *inhA* instead of the mutations of *oxyR-ahpC* because only a few studies were included to be analyzed for the latter. All the ORs were assessed using the Bayesian meta-analysis as well.

The pooled posterior value of OR for the mutations of *rpoB* (100% mutated in locus *rpoB* S531L) was [exp (log1.00)] = 2.72 ((95% confidence interval (CI): 1.90, 3.94) times higher in Beijing than in non-Beijing strains for all 31 studies included that evaluated the mutations of *rpoB*, analogous with the value of the conventional pooled OR (2.76), with a statistical significance being found in east subgroup analysis. Meanwhile, the combined heterogeneity was detected (*I*^2^ = 83.5%), and a prediction interval for the effect as [exp (log1.00)] = 2.72 (95% CI: 0.50, 15.03), meaning there would be an OR of 2.72 for the same indicators for the 32nd (*θ*_k+1_) study in the future [28] (Figure 3 and Figure 4).

The converged posterior value of OR for the mutations of *katG* (100% mutated in locus *katG S*531T) was [exp (log1.17)] = 3.22 (95% CI: 2.12, 4.90) times higher in Beijing than in non-Beijing strains for all the 27 studies included. The mutations of *katG*, comparable to the value of the pooled OR (3.26) obtained through the traditional meta-analysis, with a significant difference were found in each subgroup analysis. Simultaneously, the combined heterogeneity was detected (*I*^2^ = 90.8%), and a prediction interval for effect as [exp (log1.17)] = 3.22 (95% CI: 0.42, 24.53) was found for the 28th study in the future (Figure 5 and Figure 6).

The summarized posterior value of OR for the mutations of *inhA* (100% mutated in *inhA* -15 C > T) was [1/exp (log-0.34)] = 1.41 (95% CI: 1/exp (log0.03) = 0.97, 1/exp (log-0.73) = 2.08; the following algorithm is the same) times higher in the non-Beijing than in the Beijing strains for all 18 studies included that evaluated the mutations of *inhA*, with a significant difference found in the East/North/Central Europe group. Although the pooled posterior value of the OR between BMA and MA are close (1/0.71 vs. 1/0.70), the values of the 95% CIs of both diverted with the marginal significance, which was more obvious rather in the BMA compared to that of the MA (OR = 1.43 (95% CI: 0.95, 2.13)). Furthermore, a combined heterogeneity was detected (*I*^2^ = 63.3%), and a prediction interval for effect as 1.41 (95% CI: 0.41, 4.90) was found for the 19th study in the future (Figure 7 and Figure 8).

The pooled posterior value of OR for the mutations of *oxyR-ahpC* (100% mutated in *oxyR-ahpC* intergenic region) was [1/exp (log-0.38)] = 1.46 (95% CI: 1/exp (log0.14) = 0.87, 1/exp (log-0.91) = 2.48) times higher in the non-Beijing than in the Beijing strains for all nine studies included that evaluated the mutations of *oxyR-ahpC*, without any statistical significances found, neither in BMA nor in MA (OR = 1.45, 95% CI: 0.94, 2.22). A homogeneity (*I*^2^ = 0.0%) and a prediction interval for the effect as 1.46 (95% CI: 0.59, 3.71) were found for the seventh in a future study were identified (Figure 9 and Figure 10).

## 4. Discussion

Heterogeneities were identified by both BMA and MA in most mutations of the genes, and no publication biases were detected. Mutations of *rpoB* and *katG* related to RIF and INH were significantly more common in Beijing than in non-Beijing strains, which were not identified in the mutations of *inhA* and *oxyR-ahpC*. There was not enough evidence to demonstrate that the mutations of *inhA* and *oxyR-ahpC* were higher in the non-Beijing than in the Beijing strains.

RRDR, the so-called “hot” locus of the *rpoB* gene (81-b.p., codon 507–533) harbors around 98% of gene mutations related to RIF drug resistance [8,23]. Compared to the mutations of *katG*, which were more prevalent in European countries, combined with the evidence exhibited in the Beijing and non-Beijing strains in this study, the mutations of *rpoB* were more common in Asian countries. This is equivalent to the finding of Anwaiejiang (isolates collected in China). Despite such miscellaneous mutation locations, most of them are harbored in three *rpoB* codons: 531, 526 and 516 [9]. In this current meta-analysis, 100% of the mutations of *rpoB* were presented in the pattern of S531L. This is slightly different from a survey with isolates collected from Japan, Korea, and China [43], in which although the most prevalent mutations were similar, only *Asp*-516 was found with a higher mutation rate in Beijing than in non-Beijing isolates, different from the study results in the Kyrgyz Republic [9] and Korea [10] that displayed a lower rate in *rpoB* mutation in Beijing vs. in non-Beijing [8,65,66].

However, it might not be comprehensive to use the *rpoB* gene mutation to represent genes with mutation-conferred resistance to RIF to illustrate the drug resistance of the Beijing strain since some new variants have been found in Beijing strains. According to a previous study, the functional consequence of nonsynonymous *Rv2629* as one of the members of the *dosR* dormancy regulons was found to be upregulated under dormancy conditions in Beijing genotype strains and in a phenotype that might confer a selective advantage under microaerophilic and anaerobic conditions in Beijing strains [70].

The current review demonstrated a prevalence of 100% for the *katG*315 mutation related to INH-resistance, higher than in some previous studies [10,43,71]. The *katG* mutations in Beijing strains of MTB manifested a significantly higher rate than that in non-Beijing isolates (57.88% vs. 28.65%). The rate was higher than that of a study in Southern Xinjiang, China (30.6%; 95% CI, 25.8–35.5%, unclassified by lineages). The prevalence of the *inhA* promoter region mutation in MTB relevant to INH-resistance in Beijing was lower than that of non-Beijing strains in the East/North/Central Europe group, with a significance detected. It might be because the strains of Beijing family strains are not the predominant ones currently [22]. Notably, according to the previous study, some mutations of *inhA* are also found in drug pan-susceptible strains [71]. The drug resistance rate of *oxyR-ahpC* of Beijing strains was lower than that of non-Beijing strains (6.26% vs. 7.21%) without significance in MTB relevant to INH-resistance, although both were higher than that in a study of Isakova et al. (1.7%). Similar to the way of the *katG gene* mutation presented as *katG S315T*, almost all the mutations related to *oxyR-ahpC* happened in *the oxyR-ahpC* intergenic region (100%) [9].

This systematic review and Bayesian meta-analysis focused on combining the results of the principal gene mutations of MTB relevant to RIF/INH with the concepts of the classifications of Beijing and non-Beijing strains. It provides a snapshot of the active genes’ mutations of the circulating MTB and informs policymakers to make feasible decisions for TB control programs. Furthermore, the pooled data harbors a kind of comprehensive information that the individual study lacks, releasing the clinical practitioners with MTB genetics information for reasonable selections of the anti-TB drugs.

However, entirely relying on genetic methods is not that comprehensive and unreasonable since some potential genes’ mutations might have been discovered. Combining considerations based on the merging data of genetics, clinical and epidemiological concerns might be a promising exploration.

## 5. Limitations

There are several limitations in our study. First, due to the local practical conditions, not all the methods used in the included studies followed the World Health Organization criteria, leading to some potential heterogeneities, although corrected technically, which was not as good as it never happening. Second, although the high concern concentrations are focused on the gene mutations of anti-MTB drug resistance, not so many studies are available with the forms meeting the requirements of both the four names of gene mutation related to and the lineages of MTB as well [72], which might lead to some selection biases on the original studies interpretable for the geographical distribution. Third, we included only the major common mutations of MTB genes related to RIF and INH instead of all genes’ mutations because of the length limitation of the paper, which might not well interpret the difference in the gene mutations of the anti-TB drug resistance and the polymorphisms relevant to the two lineages of MTB.

## 6. Conclusions

The mutations in *rpoB* and *katG* genes in Beijing are significantly more common than those in non-Beijing strains of MTB. We do not have sufficient evidence to support that the prevalence of mutations of *inhA* and *oxyR-ahpC* is higher in non-Beijing than in Beijing strains, which provides a reference basis for clinical medication selection.

## Figures and Tables

**Figure 1 genes-13-01849-f001:**
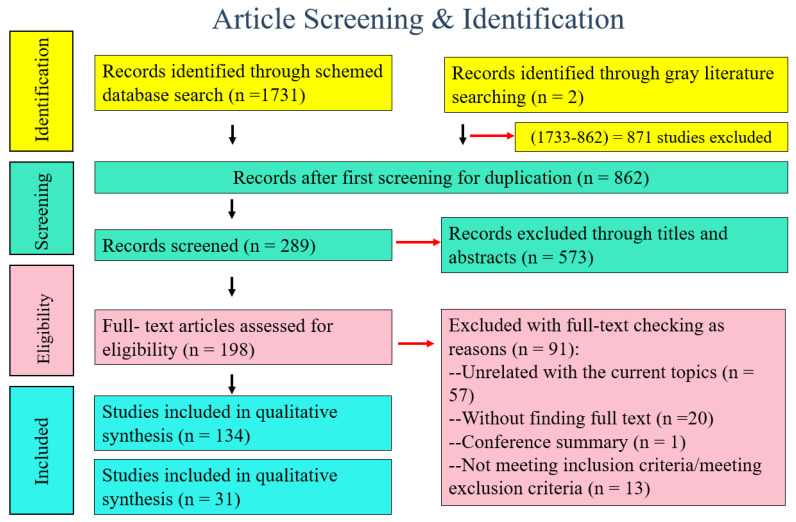
Flow diagram of literature screening and identification strategy.

**Figure 2 genes-13-01849-f002:**
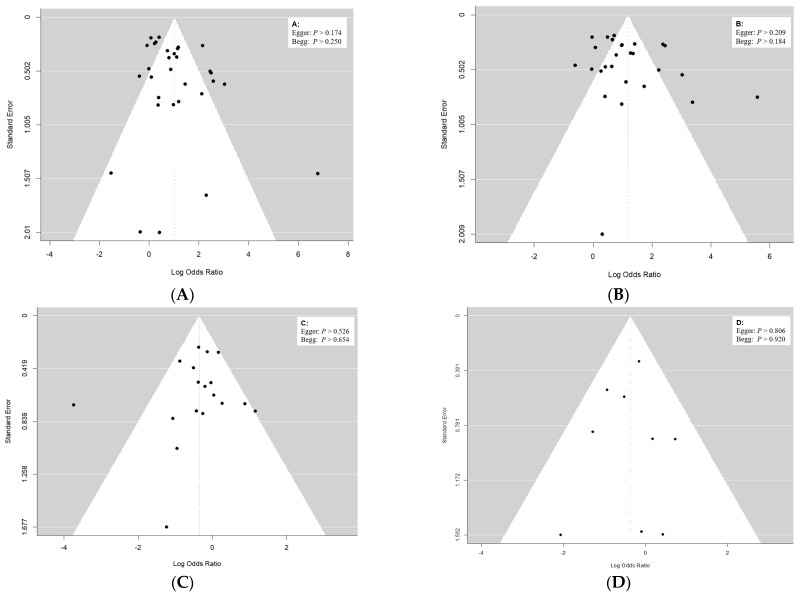
Funnel plots for publication bias tests ((**A**), *rpoB*; (**B**)*, katG*; (**C**), *inhA*; (**D**), *oxyR*−*ahpC*).

**Figure 3 genes-13-01849-f003:**
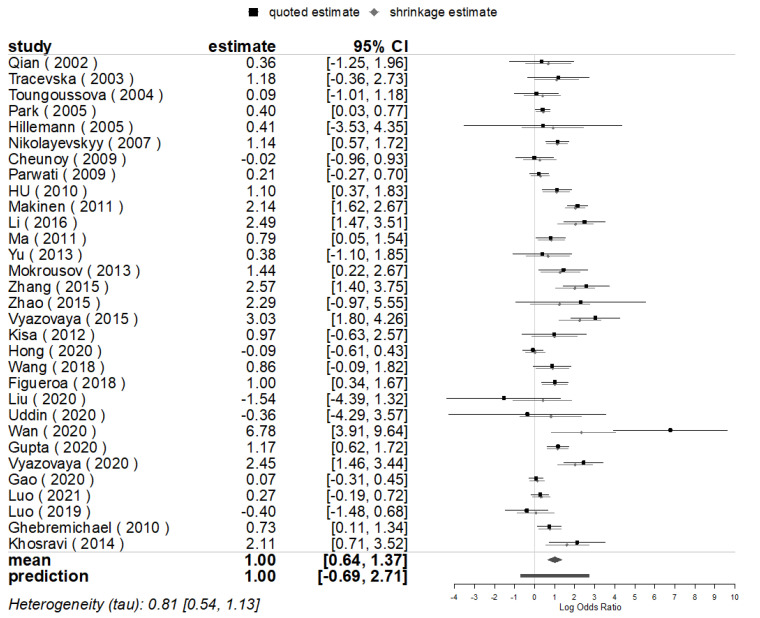
BMA for the mutations of *rpoB* in Beijing and Non-Beijing Strains (CI: confidence interval; BJ: Beijing strain; Non-BJ: non-Beijing strain).

**Figure 4 genes-13-01849-f004:**
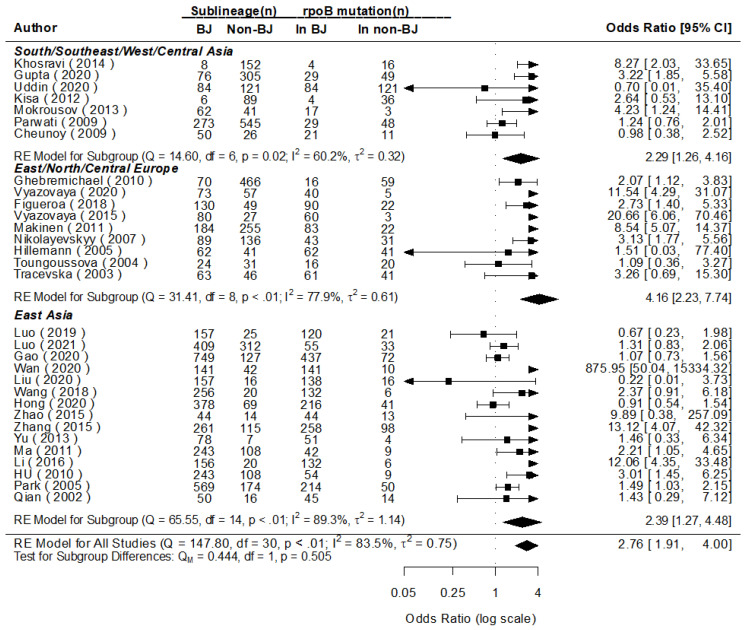
MA for the mutations of *rpoB* in Beijing and non-Beijing Strains (CI: confidence interval; BJ: Beijing strain; Non-BJ: non-Beijing strain).

**Figure 5 genes-13-01849-f005:**
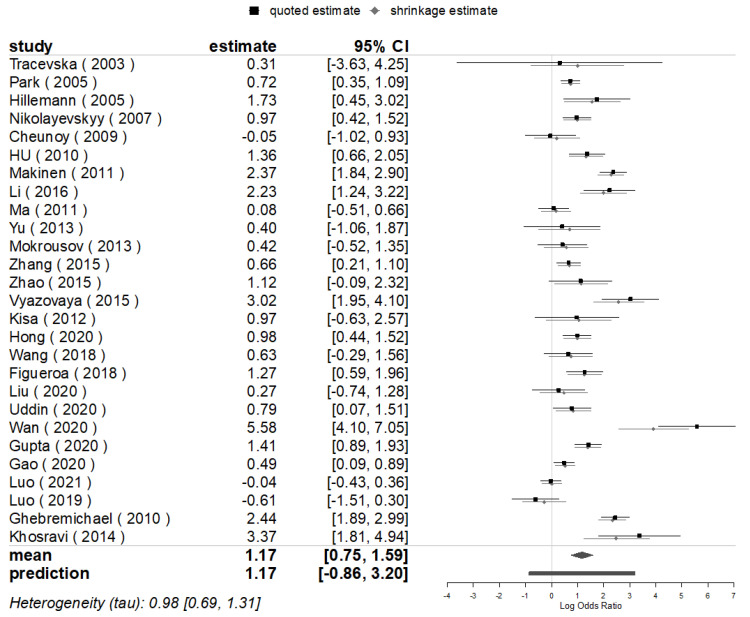
BMA for the mutations of *katG* in Beijing and non-Beijing strains.

**Figure 6 genes-13-01849-f006:**
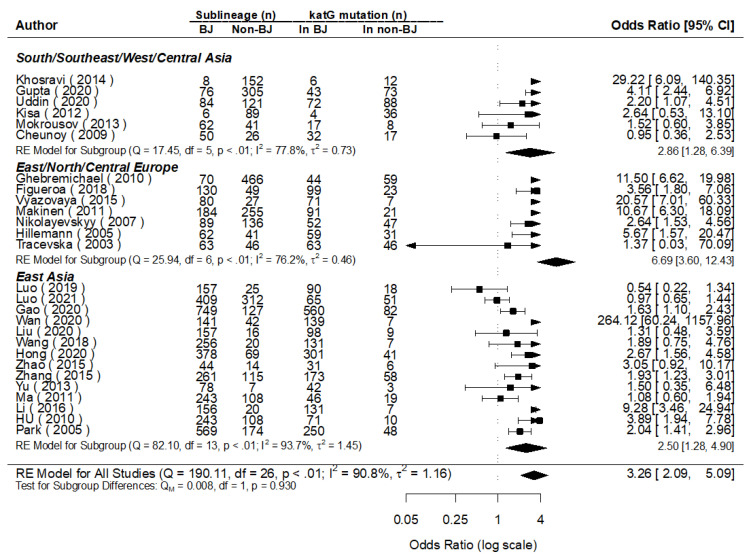
MA for the mutations of *katG* in Beijing and non-Beijing Strains (CI: confidence interval; BJ: Beijing strain; Non-BJ: non-Beijing strain).

**Figure 7 genes-13-01849-f007:**
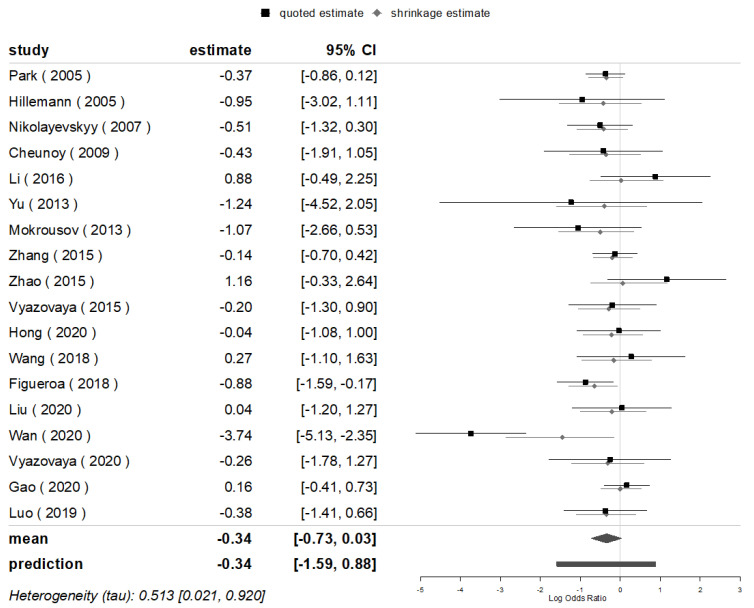
BMA for the mutations of *inhA* in Beijing and non-Beijing strains.

**Figure 8 genes-13-01849-f008:**
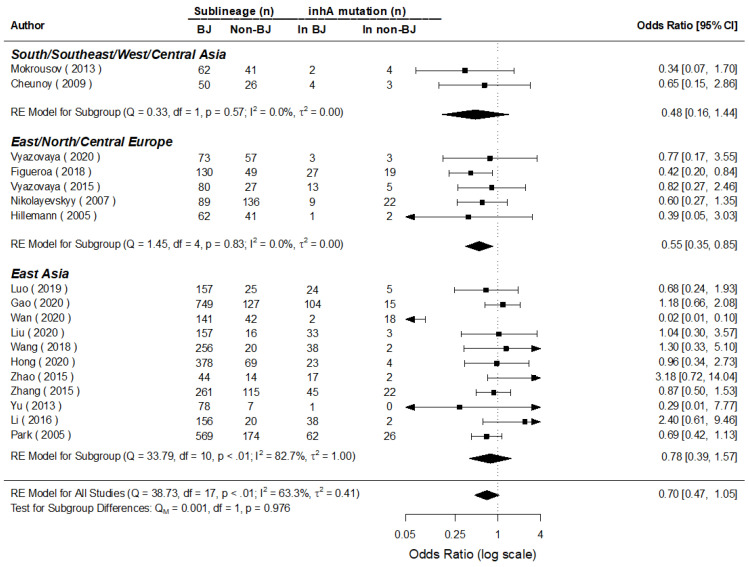
MA for the mutations of *inhA* in Beijing and non-Beijing strains (CI: confidence interval; BJ: Beijing strain; Non-BJ: non-Beijing strain).

**Figure 9 genes-13-01849-f009:**
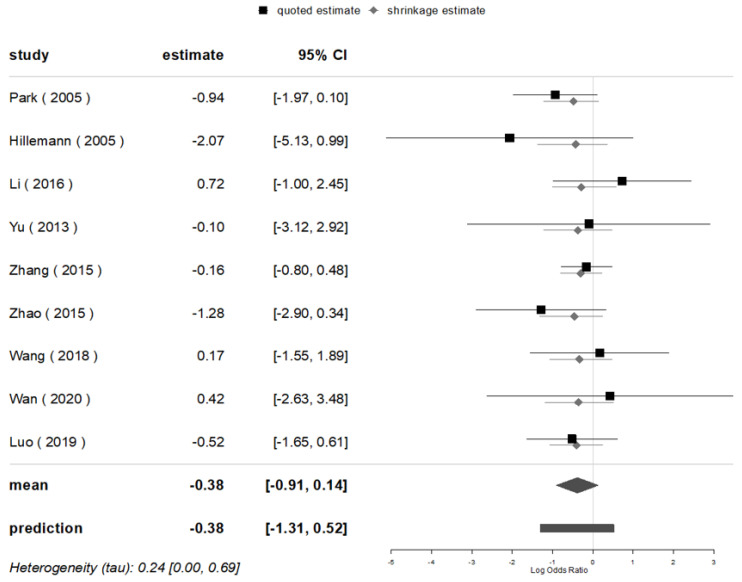
BM for the mutations of *oxyR-ahpC* in Beijing and non-Beijing strains.

**Figure 10 genes-13-01849-f010:**
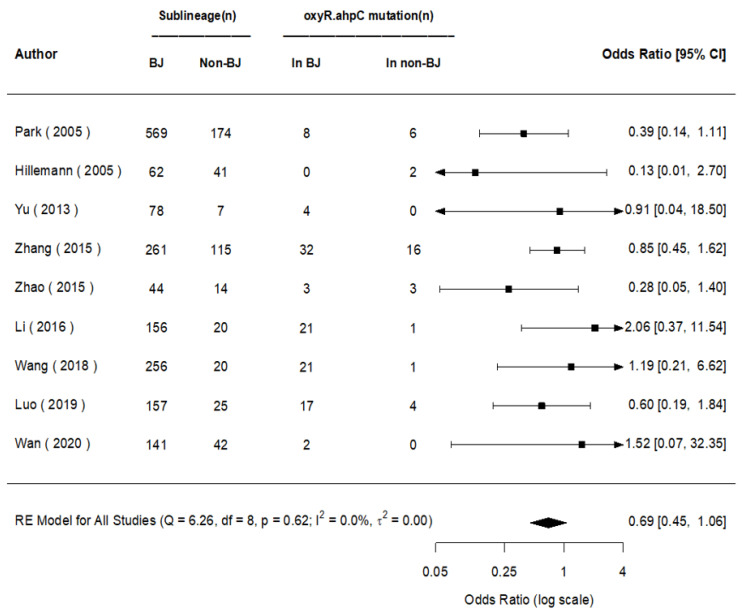
BMA&MA for the mutations of *oxyR-ahpC* in Beijing and non-Beijing strains (CI: confidence interval; BJ: Beijing strain; Non-BJ: non-Beijing strain; OxyR.C: *oxyR-ahpC*).

**Table 1 genes-13-01849-t001:** Characteristics of studies included in the systematic review and Bayesian meta-analysis.

No.	Author	Country	Year	Isolate Sample Size	Sample Size by Genotypes		*rpoB*-Rif		*katG*-INH		*inhA*-INH		*oxyR-ahpC*-INH	
					Beijing	Non-Beijing	Beijing	Non-Beijing	Beijing	Non-Beijing	Beijing	Non-Beijing	Beijing	Non-Beijing
1	Qian	Asian Countries	2002	66	50	16	45	14	NA	NA	NA	NA	NA	NA
2	Tracevska	Latvia	2003	109	63	46	61	41	63	46	NA	NA	NA	NA
3	Toungoussova	Russia	2004	55	24	31	16	20	NA	NA	NA	NA	NA	NA
4	Park	Korea	2005	743	569	174	214	50	250	48	62	26	8	6
5	Hillemann	Germany	2005	103	62	41	62	41	59	31	1	2	0	2
6	Nikolayevskyy	southern Ukraine	2007	225	89	136	43	31	52	47	9	22	NA	NA
7	Cheunoy	Thailand	2009	76	50	26	21	11	32	17	4	3	NA	NA
8	Parwati	Indonesia	2009	818	273	545	29	48	NA	NA	NA	NA	NA	NA
9	Hu	China	2010	351	243	108	54	9	71	10	NA	NA	NA	NA
10	Mäkinen	Russia	2011	439	184	255	83	22	91	21	NA	NA	NA	NA
11	Li	China	2016	176	156	20	132	6	131	7	38	2	21	1
12	Ma	China	2011	351	243	108	42	9	46	19	NA	NA	NA	NA
13	Yu	China	2013	85	78	7	51	4	42	3	1	0	4	0
14	Mokrousov	Kyrgyzstan	2013	103	62	41	17	3	17	8	2	4	NA	NA
15	Zhang	China	2015	376	261	115	258	98	173	58	45	22	32	16
16	Zhao	China	2015	58	44	14	44	13	31	6	17	2	3	3
17	Vyazovaya	Russia	2015	107	80	27	60	3	71	7	13	5	NA	NA
18	Kisa	Turkey	2012	95	6	89	4	36	4	36	NA	NA	NA	NA
19	Hong	China	2020	447	378	69	216	41	301	41	23	4	NA	NA
20	Wang	China	2018	276	256	20	132	6	131	7	38	2	21	1
21	Figueroa	Russia	2018	179	130	49	90	22	99	23	27	19	NA	NA
22	Liu	China	2020	173	157	16	138	16	98	9	33	3	NA	NA
23	Uddin	Bangladesh	2020	205	84	121	84	121	72	88	NA	NA	NA	NA
24	Wan	China	2020	183	141	42	141	10	139	7	2	18	2	0
25	Gupta	India	2020	381	76	305	29	49	43	73	NA	NA	NA	NA
26	Vyazovaya	Russia	2020	130	73	57	40	5	NA	NA	3	3	NA	NA
27	Gao	China	2020	876	749	127	437	72	560	82	104	15	NA	NA
28	Luo	China	2021	721	409	312	55	33	65	51	NA	NA	NA	NA
29	Luo	China	2019	182	157	25	120	21	90	18	24	5	17	4
30	Ghebremichael	Sweden	2010	536	70	466	16	59	44	59	8	NA	NA	NA
31	Khosravi	Iran	2014	160	8	152	4	16	6	12	NA	NA	NA	NA

**Table 2 genes-13-01849-t002:** Mutations prevalence for mutations of *rpoB*, *katG*, *inhA* and *oxyR-ahpC*.

Genes	Sample Size			Mutation Isolates		Mutation Rates		Principal Mutations Pattern
	Total	Beijing	Non-Beijing	Beijing	Non-Beijing	Beijing	Non-Beijing	
*rpoB*	8785	5225	3560	2738	930	52.40	26.12	*rpoB Ser531Leu*
*katG*	7716	4805	2911	2781	834	57.88	28.65	*katG S315T*
*inhA*	5034	3562	1472	454	157	12.75	10.67	*inhA -15 C > T, promoter region of inhA*
*oxyR-ahpC*	2182	1724	458	108	33	6.26	7.21	*oxyR-ahpC intergenic region*

## Data Availability

The data and materials that support the findings of this study are included in the article and within the Appendix A.

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
