# Peer review of "Comparison on Major Gene Mutations Related to Rifampicin and Isoniazid Resistance between Beijing and Non-Beijing Strains of Mycobacterium tuberculosis: A Systematic Review and Bayesian Meta-Analysis"

_genes, 2022, doi:10.3390/genes13101849_

Round 1

Reviewer 1 Report

Authors Guo and colleagues present a systematic review and meta-analysis of antimicrobial resistance gene mutations among M. tuberculosis isolates in Beijing and elsewhere in the world. The aim of the study was “to compare the major gene mutations related to RIF and INH resistance between Beijing and non-Beijing genotypes and extract the best evidence using evidence-based methods for improving the TB control program's service based on the genetics of MTB.”

The entire second paragraph of the introduction appears to be a duplication of all but the first sentence of the first paragraph. Furthermore, the meaning of the percents in parentheses is not clear.

Authors should include a date the search was conducted, or an end date for the literature search.

It is suggested that the authors provide as a supplement, the search terms used for each of the search engines so readers or researchers may choose to duplicate the methods.

Please clarify the inclusion criteria – criteria 1 and 3 appear mutually exclusive; “gray literature” is a term that should be defined; are all three criteria required or at least one of them?

Did the authors reach out to authors of studies with potential appropriate data that was not published or not published in its entirety (e.g., email an author for data collected as part of the study but not reported in the manuscript)?

Validity of data extraction would be more robust if they were extracted and compared with more than one investigator. “Any records with discrepancies..” and “Discrepancies between two reviewers” suggests that more than one investigator abstracted data – if that is not the case, how or where did discrepancies emerge?

In the supplemental spreadsheet, the column “Finding(s)/Appraisal(s)” seems to have incomplete data. Additionally, in other fields including Author, Country, Lab Methods, and the three mechanism fields (columns L, O, U) there also appears to be missing or incomplete data. Column L appears mislabeled (a minor point, however, reduces confidence in the veracity of data).

Author Response

Response to Reviewers’ Comments:

Reviewer 1:

Authors Guo and colleagues present a systematic review and meta-analysis of antimicrobial resistance gene mutations among M. tuberculosis isolates in Beijing and elsewhere in the world. The aim of the study was “to compare the major gene mutations related to RIF and INH resistance between Beijing and non-Beijing genotypes and extract the best evidence using evidence-based methods for improving the TB control program's service based on the genetics of MTB.”

  1. The entire second paragraph of the introduction appears to be a duplication of all but the first sentence of the first paragraph. Furthermore, the meaning of the percent in parentheses is not clear.

Per the reviewer’s advice, the second paragraph has been checked and revised, and the percent in the parentheses have been modified as shown in the first paragraph of the Introduction section.

  1. Authors should include a date the search was conducted, or an end date for the literature search.

As suggested by the reviewer, the dates of the literature have been included and displayed in the Reference section.

  1. It is suggested that the authors provide as a supplement, the search terms used for each of the search engines so readers or researchers may choose to duplicate the methods.

 According to the suggestions of the reviewer, the file called “Supplementary file 2_Search engines & terms” has been provided as another supplement and the search engines as well.

  1. Please clarify the inclusion criteria – criteria 1 and 3 appear mutually exclusive; “gray literature” is a term that should be defined; are all three criteria required or at least one of them?

 The three inclusion criteria have been modified into two points and the “gray literature” has been defined. The two revisions have been manifested in the section of 2.3 Study Selection Criteria.

  1. Did the authors reach out to authors of studies with potential appropriate data that was not published or not published in its entirety (e.g., email an author for data collected as part of the study but not reported in the manuscript)?

Yes. When we could not obtain what we expect after we had accessed the supplementary or additional files of the source literature, we reached out to some authors of studies with potential appropriate data that was not published or not published in its entirety.

  1. Validity of data extraction would be more robust if they were extracted and compared with more than one investigator. “Any records with discrepancies.” and “Discrepancies between two reviewers” suggests that more than one investigator abstracted data – if that is not the case, how or where did discrepancies emerge?

 Yes, two of us authors extracted the data and when any records with discrepancies occurred, we addressed them by referring to the source literature, and the discrepancies between the two of us were resolved by reaching a consensus involving all of us authors.

  1. In the supplemental spreadsheet, the column “Finding(s)/Appraisal(s)” seems to have incomplete data. Additionally, in other fields including Author, Country, Lab Methods, and the three mechanism fields (columns L, O, U) there also appears to be missing or incomplete data. Column L appears mislabeled (a minor point, however, reduces confidence in the veracity of data).

Per the reviewer’s advice, the supplemental spreadsheet has been improved with it being removed/added some studies and filled some appropriate data and some interpretations as well.

Reviewer 2 Report

Guo et al, compared the importance of the mutations between Beijing and Non-Beijing strains of Mycobacterium tuberculosis (MTb) in the literature to analyze the importance of those genes in the delineation and genotype of the MTb strains.  They found that the mutations prevalence of rpoB, katG, inhA and oxyR-ahpC in Beijing strains was 35.68% , 33.09%, 12.78% and 4.96%, respectively; that in non-Beijing strains was 22.64%, 20.68%, 16.26% and 6.78%, separately. These results was expected as Beijing strains have more tendency to have lots of mutations in their genome relative to Non-Beijing strains. The results is novel and interesting, however, when I checked the countries with reported Beijing and Non-Beijing strains (Table 1), there was no data from some countries of the middle east such as Iran, Afghanistan etc. which have a high burden of MTb. Authors can claim that there is no data for countries such as Afghanistan, but there is plenty of papers regarding the status of Beijing and Non-Beijing strains in Iran (for example). Adding many reports from China and ignoring the data from other countries with high burden of MTb infection will make bias in the data and underscore the quality of systematic review. I would suggest reviewers to check the availability of data for some countries in the middle east like Iran, Iraq, etc., and incorporate those data into their findings. If there is no data available (I am sure there is lots of data in the literature), then please add to the limitations of this study.

Author Response

  1. The results are novel and interesting, however, when I checked the countries with reported Beijing and Non-Beijing strains (Table 1), there was no data from some countries of the middle east such as Iran, Afghanistan etc., which have a high burden of MTb. Authors can claim that there is no data for countries such as Afghanistan, but there is plenty of papers regarding the status of Beijing and Non-Beijing strains in Iran (for example). Adding many reports from China and ignoring the data from other countries with high burden of MTb infection will make bias in the data and underscore the quality of systematic review. I would suggest reviewers to check the availability of data for some countries in the middle east like Iran, Iraq, etc., and incorporate those data into their findings. If there is no data available (I am sure there is lots of data in the literature), then please add to the limitations of this study.

Per the advice of the reviewer, we added some literature from the middle east countries and other countries, which present a high burden of MTb to reduce the selection bias of the studies included so that we reanalyzed all the data of this study and improved almost all the tables and figures as displayed int the manuscript.